# The Development of Methods of BLOTCHIP^®^-MS for Peptidome: Small Samples in Tuberous Sclerosis

**DOI:** 10.3390/cimb47010034

**Published:** 2025-01-07

**Authors:** Kunio Yui, George Imataka, Kotaro Yuge, Hitomi Sasaki, Tadashi Shiohama, Kyoichi Asada, Hidehisa Tachiki

**Affiliations:** 1Department of Pediatrics, Chiba University, Chiba-Shi 260-8677, Chiba, Japan; asuha_hare@chiba-u.jp; 2Department of Pediatrics, Dokkyo Medical University, Tochigi 321-0293, Tochigi, Japan; geo@dokkyomed.ac.jp; 3Department of Pediatrics, Kurume University, Kurume-Shi 830-0011, Fukuoka, Japan; yugekotaro@kurume-u.ac.jp; 4Department of Urology, Fujita University, Toyoake-Shi 470-1192, Aichi, Japan; sasakih@fujita-hu.ac.jp; 5Protosera Inc., Settsu-Shi 566-0002, Osaka, Japan; kasada@protosera.co.jp (K.A.); htachiki@protosera.co.jp (H.T.)

**Keywords:** tuberous sclerosis, autism spectrum disorders, pathophysiology, TSC2, glutathione

## Abstract

Mutations in TSC1 or TSC2 in axons induce tuberous sclerosis complex. Neurological manifestations mainly include epilepsy and autism spectrum disorder (ASD). ASD is the presenting symptom (25–50% of patients). ASD was observed at significantly higher frequencies in participants with TSC2 than those with TSC1 mutations. The occurrence of TSC2 mutations is about 50% larger than TSC1. Therefore, ASD may develop due to TSC2 deficiency. TSC2 regulates microRNA biogenesis and Microprocessor activity via GSK3β. Of reference, everolimus has the best treatment target because of the higher potency of interactions with mTORC2 rather than rapamycin. Mutations in the TSC1 and TSC2 genes result in the constitutive hyperactivation of the mammalian target of the rapamycin (mTOR) pathway, contributing to the growth of benign tumors or hamartomas in various organs. TSC2 mutations were associated with a more severe phenotypic spectrum than TSC1 mutations because of the inhibition of the mTOR cascade. There are few studies on the peptide analysis of this disorder in relation to everolimus. Only one study reported that, in ten plasma samples, pre-melanosome protein (PMEL) and S-adenosylmethionine (SAM) were significantly changed as diagnostic prognostic effects. Our study on peptide analysis in Protosera Inc (Osaka, Japan) revealed that three peptides that were related to inflammation in two patients with tuberous sclerosis, who showed a 30% decrease in ASD symptoms following everolimus treatment. TSC2 mutations were associated with a more severe phenotypic spectrum due to the inhibition of the mTOR cascade. PMEL and SAM were significantly changed as diagnostic effects.

## 1. Introduction

Tuberous sclerosis complex (TSC) is a genetic disease, causing benign tumors and dysfunctions of brain malformations. TSC subjects exhibit autism spectrum disorder (ASD). TSC is associated with a high prevalence of autism spectrum disorder (ASD; about 70–50% of individuals) [1]. TSC is a leading cause of syndromic ASD; understanding language development in this population would not only be important for individuals with TSC but may also have implications for those with other causes of syndromic and idiopathic ASD [2]. A better understanding of TSC-associated ASD may provide some information on the neurobiological bases of the idiopathic ASD.

## 2. TSC Pathophysiology: Study History

Tuberous sclerosis complex (TSC) is an autosomal dominant disorder characterized by the development of hamartomas in many organs, leading to ASD, epilepsy, and renal failure [3]. An inactivating mutation in two tumor-suppressor genes—TSC1 and TSC2—is the cause of this syndrome, and TSC2 mutations make up 80–90% of all mutations. A new approach for detecting mutations in TSC analysis for small TSC2 mutations, a multiplex ligation-dependent probe amplification (MLPA) analysis for large deletions in TSC1 or TSC2, and a long-range PCR/sequencing-based analysis for small TSC1 mutations are among the methods used in this research area. In a study of 65 patients with TSC, pathogenic mutations were identified in 51 patients (78%) [3]. These included thirty-six small TSC2 mutations, four large deletions involving TSC2, and eleven small TSC1 mutations. Twenty-eight of the small mutations were novel [3]. For the missense mutations, a functional assay to demonstrate that the mutations impair TSC2 protein function was found. This strategy may greatly help small- and medium-sized laboratories in the pre- and postnatal molecular diagnosis of TSC [3].

During a 17-year period (1986–2003), among 753 children with TSC (668 boys and 84 girls; a male-to-female ratio of 8:1), the prevalence of autistic disorder was 0.9%. All of these children were intellectually disabled [4].

Tuberous sclerosis complex (TSC) is an inherited disorder resulting from mutations in one of two genes, TSC1 (Hamartin) and TSC2 (Tuberin). [5]. These two proteins form a cytosolic complex that inhibits the mTOR pathway which controls cell growth and proliferation. Seizures are the most common presenting symptom. Seizures can be present in the first year of life, and up to one-third of children develop infantile spasms [5].

## 3. The Prevalence of Neuropsychiatric Disorders: Difference in the Prevalence of Neuropsychiatric Disorders Between RSC1 and TSC2

The TSC1 and TSC2 complex primarily acts as an inhibitor of the mechanistic target of rapamycin (mTOR) kinase, and mutations in either gene disrupt this pathway. The TSC/mTOR signaling axis regulates multiple anabolic and catabolic processes [6].

Important findings were reported from the large-scale international study on neuropsychiatric disorders (TAND) (tuberous sclerosis registry to raise awareness of the disease) [7]. The study enrolled 2216 eligible participants with TSC from 170 sites across 31 countries by the data cut-off for the third interim analysis (data cut-off date: 30 September 2015). The most common behavioral problems (reported in > 10% of subjects) were overactivity, sleep difficulties, impulsivity, anxiety, severe aggression, depressed mood, self-injury, and obsessions [7]. TAND included psychiatric disorders such as ASD (21.1%), attention deficit hyperactivity disorders (ADHD) (19.1%), anxiety disorders (9.7%), and depressive disorders (6.1%). Intelligence quotient (IQ) scores were available for 885 participants.Of these subjects, 44.4% had an IQ in the normal range, while mild, moderate, severe, and profound degrees of intellectual disability were observed in 28.1, 15.1, 9.3, and 3.1%, respectively [7].

## 4. The Association of Neuropsychiatric Disorders Including ASD

TAND correlations showed a higher frequency of ASD and neuropsychological deficits in TSC2 [7]. Children and those with TSC2 had significantly higher rates of intellectual disability, suggesting that age and genotype comparisons should be interpreted with caution [7] (de Vries et al., 2018). With respect to TAND and genotype, molecular testing for genetic mutations was performed in 1000 participants (45.1%). Of them, 197 had TSC1 mutations, 644 had TSC2 mutations, and 144 had no mutation identified (NMI) [7]. At the behavioral level, TSC2 mutations were associated only with self-injury at a significantly higher frequency than TSC1 (15.8% vs. 6.3%, *p* = 0.0288) [7].

At the psychiatric level, ASD was observed at significantly higher frequencies in participants with TSC2 than those with TSC1 mutations (28.6% vs. 12.2%, *p* < 0.001). ADHD, anxiety disorder, and depressive disorder were associated with TSC1 rather than TSC2 (ADHD TSC1 = 17.6%; TSC2 = 16%, *p* = 0.6881; anxiety disorder TSC1 = 10.1%; TSC2 = 8.6%; *p* = 0.7809; depressive disorders TSC1 = 10%; TSC2 = 5.2%; *p* = 0.0509) [7] (de Vries et al., 2018). Of the 93 participants with TSC1 mutations who had been evaluated using IQ-type tests, 62 (66.7%) had average intellectual ability, 15 (16.1%) had mild ID, 11 (11.8%) had moderate ID, and 5 (5.4%) had severe ID [7]. Among the 293 participants with TSC2 mutation who had been evaluated using IQ-type tests, 123 (42%) had average intellectual ability, 75 (25.6%) had mild ID, 57 (19.5%) had moderate ID, 30 (10.2%) had severe ID, and 8 (2.7%) had profound ID [7]. Significant differences were observed between TSC1 and TSC2 groups for IQ levels/categories. (*p* = 0.001) [7]. Academic/scholastic difficulties were more common in individuals with TSC2 mutations than those with TSC1 mutations (63.5% vs. 49.2%; *p* = 0.0051) [7]. TSC2 mutations tend to result in ASD, and mTOR inhibitors may be more effective, as shown in Figure 1.

More individuals with TSC2 mutations showed neuropsychological performance scores falling below the fifth percentile compared to those with TSC1 mutations (63% vs. 38.8%, *p* = 0.0024). Individuals with NMI showed IQ, academic, and neuropsychological profiles between the frequencies of the group with TSC1 and TSC2 [7].

In terms of the relationship between genotype and TAND, we observed a genotype–intellectual phenotype correlation and a higher frequency of ASD in association with TSC2 [7], a potential pattern of more depressed and anxious moods, and higher rates of anxiety and depressive disorders in association with TSC1 mutations [7].

The TOSCA registry showed a genotype–intellectual phenotype pattern, suggesting a greater likelihood of ID in participants with TSC2 than those with TSC1 mutations [7]. However, it was also important to note that only 66.7% of those with TSC1 had average intellectual ability, suggesting that a third of individuals with TSC1 may have ID [7]. Similarly, even though TSC2 mutations were more likely to be associated with ID, 42% of all individuals with TSC2 mutations had regular intellectual ability [7]. The differences between those with TSC1 and TSC2 mutations observed in other aspects of TAND were of interest, particularly as all previous genotype–phenotype studies have suggested a more “severe” phenotype associated with TSC2. The possibility that specific aspects of TAND may more likely be in association with TSC1 is therefore a potentially important observation [7].

A TSC1 vs. TSC2 differentiation may be highly oversimplistic, given that specific TSC mutations may be associated with very different functional consequences at a biochemical level [7].

With respect to ASD-associated TSC pathology, the co-occurrence of ASD and the TSC complex has been recognized for decades [8]. The features of ASD are present in 25 to 50% of individuals with TSC, potentially due to a nonspecific disruption of brain function linked to an interaction between a TSC gene and an ASD susceptibility gene [8]. Nanna D Rendtorff et al. reported the following: TSC is an autosomal dominant disorder characterized by the development of hamartomas in many organs, leading to ASD, epilepsy, and renal failure. An inactivating mutation in two tumor-suppressor genes—TSC1 and TSC2—is the cause of this syndrome, and TSC2 mutations made up 80–90% of all mutations. As a new approach for detecting mutations in TSC analysis for small TSC2 mutations, a multiplex ligation-dependent probe amplification (MLPA) analysis for large deletions in TSC1 or TSC2 and a long-range PCR/sequencing-based analysis for small TSC1 mutations were established [8]. A total of 65 patients with TSC were found, and pathogenic mutations were identified in 51 patients (78%) [3]. These included thirty-six small TSC2 mutations, four large deletions involving TSC2, and eleven small TSC1 mutations [9]. Twenty-eight of the small mutations were novel [9]. For the missense mutations, a functional assay to demonstrate that the mutations impair TSC2 protein function was found. This strategy may greatly help small- and medium-sized laboratories in the pre- and postnatal molecular diagnosis of TSC. Reference [9] reported that mutations in the TSC1 or TSC2 tumor suppressor genes lead to tuberous sclerosis complex (TSC), a dominant hamartomatous disorder that often presents with intellectual disability, epilepsy, and autism [9]. In hippocampal pyramidal neurons of mice and rats, the loss of Tsc1 or Tsc2 induces the enlargement of somas and dendritic spines, altering the properties of glutamatergic synapses. The phosphorylation of morphological changes in hippocampal pyramidal neurons was increased following the loss of Tsc2 [9]. Thus, the TSC pathway regulates growth and synapse function in neurons, and perturbations of neuronal structure and function are likely to contribute to the pathogenesis of the neurological symptoms of TSC [9]. Thus, the TSC 1 and 2 mutation in the pathophysiology of ASD was found at an earlier time.

## 5. Molecular Findings in TSC-Associated ASD

With respect to pathophysiological genetic studies on TSC-associated ASD symptoms, the following findings have been reported: an earlier report studied molecular changes associated with ASD-related symptoms such as social and cognitive deficits in the brain tissue of *Tsc1*^+/−^ mice [10].

Molecular alterations in the frontal cortex and hippocampus of *Tsc1*^+/−^ and control mice, with or without rapamycin treatment, were investigated. Thirty-three proteins which were altered in *Tsc1*^+/−^ mice were normalized following rapamycin treatment, with oxidative stress-related proteins and myelin-specific and ribosomal proteins being among them [10]. Molecular changes in the *Tsc1*^+/−^ mouse brain were more prominent in the hippocampus compared to the frontal cortex. Pathways linked to myelination and oxidative stress response were prominently affected following rapamycin treatment [10]. This study highlights potential drug targets for treating cognitive, social, and psychiatric symptoms in ASD. Similar pathways have also been implicated in other psychiatric and neurodegenerative disorders and could imply similar disease processes. Thus, the potential efficacy of mTOR inhibitors warrants further investigation not only into autism spectrum disorders, but also into other neuropsychiatric and neurodegenerative diseases [10].

With respect to TSC-associated ASD, a recent study reported the following: the mTOR pathway plays a crucial role in several brain processes leading to TSC-related epilepsy, intellectual disability, and autism spectrum disorder (ASD). The pre-natal or early post-natal diagnosis of TSC is now possible in a growing number of pre-symptomatic infants [11]. Prospective studies have highlighted that developmental trajectories in TSC infants who were later diagnosed with ASD already showed motor, visual, and social communication skills in the first year of life delays [11]. Reliable genetic, cellular, electroencephalography, and magnetic resonance imaging biomarkers can identify pre-symptomatic TSC infants at high risk of having autism and epilepsy [11]. Seizures associated with autism in a sensitive developmental time window could have the potential to mitigate autistic symptoms in infants with TSC [11].

Cell culture and mouse model experiments have identified the mTOR pathway as a therapeutic target. The decreased mean diffusivity of the right inferior cerebellar peduncle was related to a positive Autism Diagnostic Observation Schedule—Second Edition classification [12]. Studies involving diffusion tensor imaging analyses of cortico-cerebellar connectivity in relation to ASD have shown mixed results [12]. One study of a population of children with autism spectrum disorder (n = 24 males and n = 3 females, mean age 5.0 years, range = 2.6–9 years) vs. typically developing controls showed an increased mean diffusivity of bilateral superior cerebellar peduncles in the ASD group compared to the controls [12]. In a volumetric neuroimaging study of 1-year-olds with tuberous sclerosis complex, the cerebellar volume was related to neurodevelopmental severity in those with pathogenic TSC2 variants, suggesting the cerebellum diffusivity values of various white matter tracts in tuberous sclerosis complex [12]. Many children are affected by ASD and other neurodevelopmental disorders, and treatment trials offer the hope of these child having more effective communication or enhanced social skills. Recent evidence from mouse models indicate that even disorders that appear quite similar in terms of cell biology (i.e., regulation of protein synthesis) such as TSC and FXS may have diametrically opposite physiological phenotypes under certain circumstances [12]. Therefore, a detailed understanding of the circuitry and cellular abnormalities associated with each single-gene disorder is crucial in selecting the most effective treatment [12].

At the following stage, cell culture and mouse model experiments have identified the mTOR pathway as a therapeutic target. Studies involving the diffusion tensor imaging analysis of cortico-cerebellar connectivity in relation to ASD have led to mixed results [12].

## 6. Differences in Molecular Mechanisms Between TSC1 and TSC2

TSC2 mutations were correlated with a more severe phenotypic spectrum than TSC1 mutations. The deletion of exon 4 in TSC2 affected cell proliferation, migration, and cell cycle via the abnormal activation of the PAM signaling pathway [13]. Mutations in the TSC2 gene are likely to have dominant effects on social behaviors rather than the TSC1 gene. Understanding the genotype–phenotype relationship may contribute to the development of ameliorate social deficits in ASD.

The use of different beta-cell lines, both those expressing the insulin receptor and those which do not (IR(+/+) and IR(−/−)) or those with a reconstituted expression of IR isoforms, revealed that both phosphatidylinositol 3-kinase/Akt/TSC/mTOR complex 1 and MAPK kinase/ERK pathways mediate insulin signaling in IR(+/+)-, IRA-, or IRB-expressing cells [14]. However, glucose signaling was mediated by MAPK kinase/ERK and AMP-activated protein kinase pathways, as assessed in IR(−/−) cells [14]. The knockdown of TSC2 expression upregulated the downstream basal phosphorylation of 70 kDa ribosomal protein S6 kinase (p70S6K) and mTOR [14]. The regulation of TSC2 phosphorylation by insulin or glucose independently integrates beta-cell proliferation signaling, with the relative expression of IRA or IRB isoforms in pancreatic beta cells playing a major role [14].

TSC1 and TSC2 genes were found in 53 patients with high suspicion of TSC. The confirmation of all variants was conducted by the Sanger method. TSC2 mutations were associated with a more severe phenotypic spectrum than TSC1 mutations. TSC2 mutation representation is about 50% larger than TSC1. TSC1 contains more repeat elements than TSC2 (32% vs. 25% total sequence) [15]. TSC2 rearrangements were seen in our cohort, while TSC1 rearrangements were not. Mutations were distributed throughout all gene regions with the exception of the 3’ region of TSC2. There was a high occurrence of splice site mutations at the donor site of exon 10 in TSC2. Importantly, TSC2 variants were associated with a more severe phenotypic spectrum, as compared to TSC1 variants [15].

With respect to the molecular analyses of TSC1, a recent study indicated the novel finding that murine postnatal subventricular zone (SVZ) neural stem progenitor cells (NSPCs) were deficient in Tsc1 gene. 2D-DIGE-based proteomic analysis detected 55 differently represented spots in Tsc1-deficient cells, as compared to wild-type counterparts, which were associated with 36 protein entries after corresponding trypsinolysis and nano-LC-ESI-Q-Orbitrap-MS/MS analysis [16]. Changes in the abundance of proteins involved in oxidative/nitrosative stress, cytoskeleton remodeling, neurotransmission, neurogenesis, and carbohydrate metabolism were found [16]. These proteins reveal novel molecular aspects of TSC etiopathogenesis and may serve as potential molecular targets for managing TSC-related disorders [16].

In the global impact of TSC2 on microRNAs, an old study quantitatively analyzed 752 microRNAs in Tsc2-expressing and Tsc2-deficient cells [17]. Out of 259 microRNAs expressed in both cell lines, 137 were significantly upregulated and 24 were significantly downregulated in Tsc2-deficient cells, consistent with the increased Microprocessor activity. Microprocessor activity is known to be regulated in part by GSK3β. Tsc2-deficient cells and the increase in Microprocessor activity associated with Tsc2 loss were reversed by three different GSK3β inhibitors [17]. mTOR inhibition increased the levels of phospho-GSK3β (S9), which negatively affects Microprocessor activity. Taken together, these data reveal that TSC2 regulates microRNA biogenesis and Microprocessor activity via GSK3β [17].

The pathogenicity of the large heterozygous deletion of exon 4 in TSC2 in a three-generation TSC family was analyzed via whole exome sequencing. The deletion of exon 4 in TSC2 affected cell proliferation, migration, and cell cycle via the abnormal activation of the PAM pathway. Thus, the deletion of exon 4 in TSC2 had a pathogenic effect [18].

## 7. mTOR Activity in Tuberous Sclerosis

Cells respond to DNA damage by activating a complex array of signaling networks, which include the adenosine monophosphate active protein (AMPK) and mTOR pathways. After DNA double-strand breakage, ATM, a core component of the DNA repair system, activates the AMPK-TSC2 pathway, leading to the inhibition of the mTOR cascade [19].

The nervous system is generated during a relatively short period of intense neurogenesis that is orchestrated by a number of key molecular signaling pathways [20]. TSC is caused by mutations in the TSC1 or TSC2 gene, leading to the activation of the mechanistic target of the rapamycin (mTOR) signaling pathway [20]. TSC neurobiology and how the use of animal model systems has provided insights into the roles of mTOR signaling in neuronal differentiation and migration were both areas of interest. Continuing efforts to understand mTOR neurobiology will help to identify new therapeutic targets for TSC and other neurological diseases [20].

With respect to mTOR pathway activation in TSC, genetic epistasis analysis positioned both dTSC1 and dTSC2 downstream of the insulin-phosphatide inositol-3-kinase (PI3K) pathway and upstream of ribosomal protein S6 kinase 1 (S6K1) [20]. TSC1/TSC2 was found to repress mTOR complex 1 (mTORC1) to signal to downstream substrates. Given that Rheb-GTP is sufficient to activate mTORC1 [20], TSC2 is inhibited through direct phosphorylation by ribosomal S6 kinase (RSK) within the MAPK signaling pathway. TSC1/2 at the converging point of two major signaling pathways, PI3K/Akt and MAPK/RSK, can control cell growth through mTORC1 [20].

Mutations in the TSC1 and TSC2 genes result in the constitutive hyperactivation of the mammalian target of the rapamycin (mTOR) pathway, contributing to the growth of benign tumors or hamartomas in various organs. Due to the implication of mTOR pathway dysregulation in the disease pathology, increasing evidence supports the use of mTOR inhibitors for treating multiple manifestations of TSC [21].

In summary, cells in the nervous system received DNA damage by the activation of signaling networks, such as the AMPK and mTOR pathways. After DNA double-strand breakage, the AMPK-TSC2 pathway is activated, leading to the inhibition of the mTOR cascade.

## 8. Medical Treatment in Tuberous Sclerosis

The loss of function mutations in TSC1 and TSC2 genes by the aberrant activation of the mechanistic target of rapamycin (mTORC1) signaling pathway can cause TSC. Although the mTORC1 inhibitor rapamycin has demonstrated exciting results, the tumors did not respond [22]. Thus, the identification of additional molecular targets and the development of more effective remission-inducing therapeutic strategies are necessary for TSC patients [23]. Everolimus has a higher potency of interacting with mTORC 2 than rapamycin. Everolimus demonstrated better ability than rapamycin in treating subependymal giant cell astrocytoma and other tuberous sclerosis manifestations, based on more robust clinical trial experience [22]. Everolimus has demonstrated better efficacy than rapamycin in treating subependymal giant cell astrocytoma and other tuberous sclerosis manifestations [23].

Mechanistically, mTORC1 inhibition increased the mitogen-activated protein kinase 1 (MEK1)-dependent activation of a mitogen-activated protein kinase (MAPK) in TSC-deficient cells [23]. The combinatorial inhibition of mTORC1 and MAPK induces the death of TSC2-deficient cells. The mTOR1 pathway is the main therapeutic target for TSC patients. Long-term rapamycin treatment decreases mTORC2 signaling in primary human dermal microvascular endothelial cells and several cell lines [23]. Given the impact of rapamycin on the pro-survival of TSC mutant cells, the effect of everolimus on the survival of TSC mutant cells requires further investigation [23]. mTORC1 inhibition leads to the upregulation of pro-survival mediators, including autophagy, and paradoxically increases the growth of Tsc2-null cells [23]. mTORC1 inhibition using rapamycin results in a compensatory activation of MAPK in TSC1- and TSC2-deficient cells [23]. This enhanced MAPK signaling pathway was associated with the enhanced survival of TSC-deficient cells in vitro [23]. The dual inhibition of mTORC1 and MAPK triggers the death of TSC2-deficient cells [23].

The mechanistic target of rapamycin (mTOR) kinase mediates various long-lasting forms of synaptic and behavioral plasticity [24]. To indicate the effects of a single injection of the dual mTROC1/2 inhibitor AZD2014 after learning from memory consolidation and persistence, a dose–response experiment was carried out. A single systemic injection of rapamycin impaired the formation and persistence of contextual fear memory, but the results require the further understanding of the role of mTORC1/2 kinase activity in the molecular mechanisms underlying memory processing [24].

Additionally, we must also focus on the developmental and epileptic encephalopathies characterized by severe drug-resistant epilepsy and significant neurodevelopmental comorbidities, which include ASD symptoms and psychiatric problems including anxiety and depression, speech impairment, and sleep problems (Str). The mainstay of treatment involves multiple anti-seizure medications (ASMs), which have effects on cognition. The ASMs include valproate (VPA), clobazam, topiramate (TPM), cannabidiol (CBD), fenfluramine (FFA), levetiracetam (LEV), brivaracetam (BRV), zonisamide (ZNS), perampanel (PER), ethosuximide, stiripentol, lamotrigine (LTG), rufinamide, vigabatrin, lacosamide (LCM), and everolimus. CBD, FFA, LEV, BRV, and LTG may have some positive effects [25]. LTG is associated with insomnia. Treatment regimes are complex, involving multiple ASMs as well as other drugs [25].

The pathogenic molecular mechanisms of ASD consist of the hyperactivation of mTORC1, high mTORC1 activity in the brain, and ASD-related behavioral deficits, which were reversed by the mTORC1 inhibitor rapamycin. Environmental stress may affect this signaling pathway. mTORC1 is a signaling molecule for diverse cellular functions, including protein synthesis. The effects of mTORC1 inhibitors [26] are significant. 

According to a recent review article, mTORC1 consists of raptor (regulatory associated protein of mTOR), whereas mTORC2 consists of rictor (rapamycin-insensitive companion of mTOR) [26]. The signal mediators are transmitted to TSC1/2, which is suppressed by inputs from most pathways by the AMPK pathway. The activation of TSC1/2 suppresses downstream Rheb. Rheb directly stimulates mTORC1 [26]. Several human genetic disorders are known to be associated with mTORC1 activation and ASD [26]. Rapamycin does not directly bind to mTOR but instead binds to FKBP12 (FK506-binding protein 12) to form the FKBP12–rapamycin complex, bind to mTOR in mTORC1, and suppress mTORC1 activity, while mTORC2 does not interact with the FKBP12 complex [26]. Rapamycin and its analogs are expected to reverse mTORC1 hyperactivity and related neuropsychiatric deficits, such as ASD. mTORC1 inhibitors are the most promising drugs that can improve ASD [26]. The efficacy of mTORC1 inhibitors and other promising agents, such as mGlu5 inhibitors, in ameliorating ASD requires further studies are required [26].

## 9. Peptide Analysis: Everolimus Related Analysis

There were several studies reporting changes in plasma peptide in relation to the everolimus treatment.

A previous study conducted the profiling of the plasma proteomics and metabolomics of patients with renal cysts, sporadic angiomyolipoma (S-AML), and tuberous sclerosis complex-related angiomyolipoma (TSC-RAML) both before and after everolimus treatment [27]. The plasma proteomics and metabolomics of patients with renal cysts, sporadic angiomyolipoma (S-AML), and tuberous sclerosis complex-related angiomyolipoma (TSC-RAML) were analyzed to identify diagnostic and prognostic biomarkers, revealing the underlying mechanisms of TSC tumorigenesis, such as pre-melanosome protein (PMEL) and S-adenosylmethionine (SAM) for diagnostic and prognostic effects. S-adenosylmethionine (SAM) demonstrated both diagnostic and prognostic effect [27].

A recent study was conducted to obtain the profile of the plasma proteomics and of patients with renal cysts, sporadic angiomyolipoma (S-AML), and tuberous sclerosis complex-related angiomyolipoma before and after everolimus treatment, and also to obtain potential diagnostic and prognostic biomarkers of TSC tumorigenesis [27]. The tumor reduction rates of TSC-RAML were assessed and correlated with the plasma levels of the proteins and metabolites. Further, a functional analysis based on differentially expressed molecules was performed to reveal the underlying mechanisms [27].

Eighty-five patients with one hundred and ten plasma samples were enrolled in this study. Multiple proteins and metabolites, such as pre-melanosome protein (PMEL) and S-adenosylmethionine (SAM), were significantly changed as diagnostic and prognostic effects [27]. These results were useful to clarify some molecular aspects of TSC etiopathogenesis and novel therapeutic protein targets. Murine postnatal subventricular zone (SVZ) neural stem progenitor cells (NSPCs) deficient of the Tsc1 gene were used as a model of the disease. Four plasma proteomics and the metabolomics pattern of TSC-RAML were significantly different from that of other renal tumors, and the differentially expressed molecules could be used as prognostic and diagnostic biomarkers [27]. Thus, this abundance of proteins are involved in oxidative/nitrosative stress, cytoskeleton remodeling, neurotransmission, neurogenesis, and carbohydrate metabolism.

## 10. Our Peptide Analysis in Small Samples

As a few studies present changes in plasma peptide, our recent study revealed novel peptides in small samples. 

Serum peptidomic analysis was performed, along with BLOTCHIP^®^-MS analysis. Pretreatment with 4–12% gradient sodium dodecyl sulfate (SDS) polyacrylamide gel electrophoresis effectively separated peptides from serum proteins [28].

Following separation, peptides were electroblotted onto BLOTCHIP^®^ (Protosera Inc., Osaka, Japan). The MALDI matrix, α-cyano-4-hydroxycinnamic acid (CHCA) (Sigma-Aldrich Co., St. Louis, MO, USA), was directly applied onto BLOTCHIP^®^, and peptidome profiles were acquired in a linear mode using ultrafleXtreme TOF/TOF (Bruker Inc., Billerica, MA, USA). All measurements were repeated four times. Statistical analyses of MS spectral data were performed using ClinProTools v3.0 (Bruker Inc., Billerica, MA, USA). The software facilitated baseline subtraction, normalization, recalibration, and the peak picking of the acquired MS spectra. Peak heights with significant differences (*p* < 0.05) between the two groups were analyzed using the Wilcoxon signed-rank test, a nonparametric test suitable for paired comparisons.

As shown in Table 1, two participants demonstrated improvement. Case 1 exhibited a 30% decrease in the ADI-R A score, and Case 2 showed improvement in all three ADI-R domains (ADI-R A; ADI-R B; and ADIR-C) and the ABC total score.

BLOTCHIP^®^-MS analysis indicated that the serum concentration of certain peptides with *m*/*z* values of 1045, 2789, and 3107 increased or decreased in the two subjects who showed 30% decreases in the behavioral symptom improvement with everolimus treatment. Peptide 1045 is known to be an important biomarker for monitoring changes in high-risk status for infant caries. Dental caries may be related to neuroinflammation, suggesting a potential connection to brain function [28].

Peptides at 2789 *m*/*z* showed a significant correlation with serum ferritin levels [29]. The relationship between ferritin and the neurodegenerative diseases was reported to be linked to neurodegenerative disorders, including ASD [30]. The signals detected accurate masses of *m*/*z* 3107 in the uptake and metabolization of the sartan drug telmisartan by a series of plants [31]. Furthermore, 1045 *m*/*z* peptides potentially related to plasma or tissue extracts in Angiotensin I or II were monitored with mass spectroscopy [32].

The present findings on 3107 suggest a potential link with neurodegenerative disorders, such as ASD. In summary, *m*/*z* might be relevant to TSC-associated ASD. Further investigation is warranted to elucidate the specific roles of these peptides in everolimus therapy or TSC pathology.

## 11. Summary

Tuberous sclerosis complex (TSC) is an inherited disorder resulting from the mutations of two genes, TSC1 (Hamartin) and TSC2 (Tuberin). These two proteins form a cytosolic complex, inhibiting the mTOR pathway that controls cell growth and proliferation. Namely, a loss of functional mutations in TSC1 and TSC2 genes by the activation of the mechanistic target of the rapamycin (mTORC1) signaling pathway caused TSC. ASD is the most common presenting symptom, being observed in up to one third of children with ASD (25–50% of the patients). ASD was observed at a significantly higher frequency in participants with TSC2 than those with TSC1 mutations, while ADHD, anxiety disorder, and depressive disorder were more associated with TSC1 rather than TSC2, indicating a more severe neuropsychiatric phenotype being associated with TSC2. The number of hippocampal pyramidal neurons increased following Tsc2 loss. In contrast, molecular changes in the Tsc1^+/−^ mouse brain were more prominent in the hippocampus.

TSC2 mutations were associated with a more severe phenotypic spectrum than TSC1 mutations. TSC2 mutations represent about 50% more cases than TSC1. Therefore, ASD may develop due to the TSC2 deficiency. TSC2 regulates microRNA biogenesis and Microprocessor activity via GSK3β. Tsc2 loss was reversed by three different GSK3β inhibitors. mTOR inhibition increased the levels of phospho-GSK3β (S9). TSC2 regulates microRNA biogenesis and Microprocessor activity via GSK3β, and a core component of the DNA repair system activates the AMPK-TSC2 pathway, leading to the inhibition of the mTOR cascade.

Signaling pathways linked to both myelination and oxidative stress representation were prominently affected following the medical target of rapamycin. Although the mTORC1 inhibitor rapamycin has demonstrated exciting results, the tumors did not respond. Of reference, everolimus has the best treatment target because of the higher potency of interactions with mTORC 2 than rapamycin. Mutations in the TSC1 and TSC2 genes result in the constitutive hyperactivation of the mammalian target of the rapamycin (mTOR) pathway, contributing to the growth of benign tumors or hamartomas in various organs.

Conclusively, TSC2 mutations were associated with a more severe phenotypic spectrum than TSC1 mutations because of the inhibition of the mTOR cascade.

There are a few studies on the peptide analysis of tuberous sclerosis disorders. Only one study revealed that among eighty-five patients with one hundred and ten plasma samples, pre-melanosome proto (PMEL) and S-adenosylmethionine (SAM) were significantly changed as diagnostic and prognostic effects. Our recent peptide analysis in Protosera Inc, Japan, revealed that the two peptides showing a 30% decrease in ASD symptoms following everolimus treatment were related to inflammation.

Our peptide analyses indicated that only *m*/*z* 3107 may be a potential link to neurodegenerative disorders, such as ASD.

## Figures and Tables

**Figure 1 cimb-47-00034-f001:**
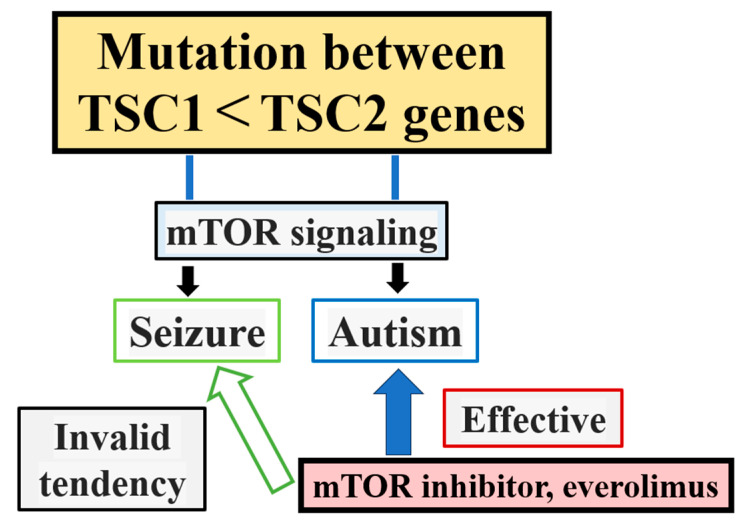
ASD is found more frequently in SC2 mutations than in TSC1 mutations.

**Table 1 cimb-47-00034-t001:** Clinical characteristics in the seven cases.

Case NO Ages	Diagnosis Major Features	Minor Features	Clinical Symptoms	Deceased Percent of Symptoms
Case 1	Subpendymal nodule	None	Restricted interest	ADI-RA 36.0
Male	Cortical tuber		Stereotyped behavior	ADI-R B 28.5
6 years	Cardiac rhabdomyoma		Deficit communication	ADI-R C 27.0
			Epilepsy	ABC 22.0
				SRS 12.9
Case 2	Subpendymal nodule	Confetti skin lesions	Restricted interest	ADI-R A 44.0
Female	Cortical tuber		Deficit communication	ADI-R B 69.0
18 years	Cardiac rhabdomyoma		Epilepsy	ADI-R C 100.0
				ABC 76.0
				SRS 13.0
Case 3	Subpendymal nodule	Confetti skin lesions	Persistent deficit in	ADI-R A Worsen
Male	Cortical tuber		social interaction	ADI-R B 8
14 years	Astrocytoma		Intellectual disability	ADI-R C 27
				ABC Worsen
				SRS 10.0
Case 4	Subependymal giant cell	None	Difficulties in	ADI-R A 5
Male	Renal angiomyolipomas		social interaction	ADI-R B 0
16 years			Moderate Intellectual	ADI-R C 0
			Disability	ABC 16
				SRS 4
Case 5	Subependymal giant cell	None	Difficulties in	ADI-R A 5
Male	Renal angiomyolipomas		social interaction	ADI-R B 0
17 years			Restricted, repetitive, and sensory interest	ADI-R C 0
			Severe intellectual disability	ABC 0
				SRS 2
Case 6	Subependymal giant cell	None	Difficulties in	ADI-R A 4
Female	Renal angiomyolipomas		social interaction	ADI-R B 0
12 years			Restricted, repetitive, and sensory interest	ADI-R C 0
			Mild intellectual disability	ABC 9
				SRS 3
Case 7	Subependymal giant cell	None	Difficulties in verbal	ADI-R A 0
Male	Renal angiomyolipomas		social interaction	ADI-R B 0
11 years			Restricted interest	ADI-R C 0
			Hyperactivity	ABC 1
			Difficulties adjusting behaviors to social situations (Irritability)	SRS 11

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
