# Peer review of "The Development of Methods of BLOTCHIP®-MS for Peptidome: Small Samples in Tuberous Sclerosis"

_cimb, 2025, doi:10.3390/cimb47010034_

Round 1
Reviewer 1 Report
Comments and Suggestions for Authors
Dear authors,
Since the manuscript format is not article and there are no data, I highly recommend authors submit it to the journal as case report and review.
Author Response
Reviewer 1
Dear authors,
Since the manuscript format is not article and there are no data, I highly recommend authors submit it to the journal as case report and review.
Our Answer
First, we submitted four case reports on peptide analysis using BLOTCHIP®-MS to “biomedecines / MDPI”. However, the article was not peer-reviewed. After consulting with Mr. Tang, Editorial Manager, he suggested that we submit a review article with our peptide analysis to the Tuberous sclerosis associated review article the autism spectrum disorders. We re-titled the article: “Development of a peptidome analysis method in autism spectrum disorders associated with tuberous sclerosis”.
Would you accept our intention to submit it as a review article?

Reviewer 2 Report
Comments and Suggestions for Authors
Kunio et al. Mutations in the TSC2 gene are more likely to lead to autism spectrum disorder (ASD) than TSC1, everolimus improves ASD symptoms in TSC patients by inhibiting the mTOR pathway, and PMEL and SAM may be potential diagnostic markers. Moreover, the result is technically sounded and worthy to be published in CURRENT ISSUES IN MOLECULAR BIOLOGY.
The following are some comments and suggestions that are given to improve the manuscript:
Comment 1: TSC2 mutations are more common than TSC1 mutations, and ASD is more frequent in patients with TSC2 mutations, whether this means that TSC2 mutations are more likely to cause ASD.
Comment 2: The article mentions that everolimus interacts more effectively with mTORC2 than Rapamycin, does this explain its advantage in the treatment of TSC-related ASD?
Comment 3: PMEL and SAM are significantly altered in the plasma of TSC patients, and is this related to the pathophysiological mechanism of TSC?
Comment 4: In addition to everolimus, there are drugs or treatments that target the mTOR pathway to treat TSC-related ASD.
Author Response
Reviewer 2
Comments and Suggestions for Authors:
Kunio et al. Mutations in the TSC2 gene are more likely to lead to autism spectrum disorder (ASD) than TSC1, everolimus improves ASD symptoms in TSC patients by inhibiting the mTOR pathway, and PMEL and SAM may be potential diagnostic markers. Moreover, the result is technically sounded and worthy to be published in CURRENT ISSUES IN MOLECULAR BIOLOGY.
The following are some comments and suggestions that are given to improve the manuscript:
Comment 1: TSC2 mutations are more common than TSC1 mutations, and ASD is more frequent in patients with TSC2 mutations, whether this means that TSC2 mutations are more likely to cause ASD.
Our answer
We added sentences that TSC2 mutations are more common than TSC1 that mutations, and ASD is more frequent in patients with TSC2 mutations, whether this means that TSC2 mutations are more likely to cause ASD (Kashii H et al. 2023) [13] in the section 6 as mark of green color.
Comment 2: The article mentions that everolimus interacts more effectively with mTORC2 than Rapamycin, does this explain its advantage in the treatment of TSC-related ASD?
Our answer
Summarizing findings indicated that everolimus demonstrated better ability than rapamycin treating subependymal giant cell astrocytoma and other tuberous sclerosis manifestations (Lu et al. 2020) [23] in the section 8 marked by green color.
Comment 3: PMEL and SAM are significantly altered in the plasma of TSC patients, and is this related to the pathophysiological mechanism of TSC?
Our answer
We added the following sentences on PMEL and SAM in the section 9, marked by green color
Comment 4: In addition to everolimus, there are drugs or treatments that target the mTOR pathway to treat TSC-related ASD.
Our answer
We added the sentences on the effect of rapamycin and anti-seizure treatment in the section 8 (Medical treatment in tuberous sclerosis) marker by green color.

Round 2
Reviewer 1 Report
Comments and Suggestions for Authors
Dear authors,
OK. I understand and accept your intention.
Reviewer 2 Report
Comments and Suggestions for Authors
The authors have answered all my previous questions.